# Unsupervised Hierarchical Graph Representation Learning with Variational Bayes

## Abstract

Hierarchical graph representation learning is an emerging subject owing to the increasingly popular adoption of graph neural networks in machine learning and applications. Loosely speaking, work under this umbrella falls into two categories: (a) use a predefined graph hierarchy to perform pooling; and (b) learn the hierarchy for a given graph through differentiable parameterization of the coarsening process. These approaches are supervised; a predictive task with ground-truth labels is used to drive the learning. In this work, we propose an unsupervised approach, BAYESPOOL, with the use of variational Bayes. It produces graph representations given a predefined hierarchy. Rather than relying on labels, the training signal comes from the evidence lower bound of encoding a graph and decoding the subsequent one in the hierarchy. Node features are treated latent in this variational machinery, so that they are produced as a byproduct and are used in downstream tasks. We demonstrate a comprehensive set of experiments to show the usefulness of the learned representation in the context of graph classification.

## 1 Introduction

Graph representation learning has attracted a surge of interest recently, inspired by the widespread success of representation learning in the image and language domains through the use of deep neural networks for parameterization. A substantial number of graph neural network (GNN) architectures (Bruna et al., 2014; Henaff et al., 2015; Duvenaud et al., 2015; Defferrard et al., 2016; Kipf & Welling, 2017; Hamilton et al., 2017; Chen et al., 2018; Veličković et al., 2018; Ying et al., 2018a; Liao et al., 2019; Xu et al., 2019) extend the convolution filters for a regular grid of data (e.g., image pixels, time series, and sequences) to irregularly connected graph neighborhoods. This extension naturally stimulates the quest of also extending the pooling operation in convolutional neural networks (CNN) to graphs. The challenge lies in the irregular connections as opposed to a regular grid structure, whereby partitioning is straightforward.

Graph pooling is used in at least two scenarios. One is global pooling: it pools the vector representations of all nodes to a single vector as the graph representation. Simple operators such as max or mean are applied. A slightly more complex operator is the weighted sum, wherein the weights are computed through attention (Veličković et al., 2018; Lee et al., 2019). A recently proposed operator is top-k pooling (Zhang et al., 2018), whereby a fixed number of node representations at the top of a sorted list is retained so that convolutions or feed-forward transformations are applied.

The second use of pooling is the creation of a graph hierarchy. In this scenario, pooling is local, more similar to that in CNNs. It is interfaced with graph coarsening (also called graph reduction or graph compression), a generic form of which is to cluster the nodes of the original graph into a node of the coarse graph. Then, the representations of the nodes in the cluster are pooled. The clustering may be obtained by using existing graph coarsening or graph clustering approaches, as in Bruna et al. (2014); Defferrard et al. (2016); Simonovsky & Komodakis (2017); or learned through parameterization as in Ying et al. (2018b); Gao & Ji (2019); Lee et al. (2019). In either approach, the result include both a hierarchy of graphs and the accompanying node representations.

Representation learning in these local pooling approaches is supervised, with the training signal coming from labels of the downstream task. In this work, we propose an unsupervised learning approach, named BAYESPOOL, through the use of variational Bayes. We use an existing coarsening

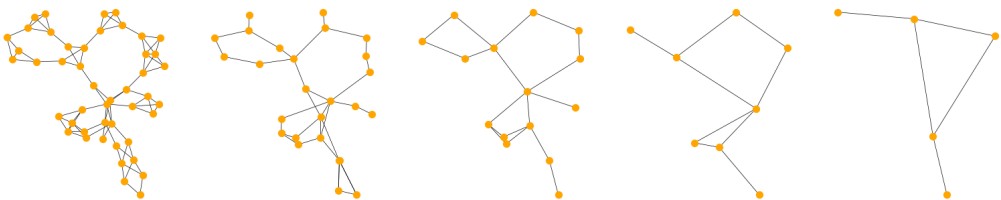

Figure 1: Coarsening sequence of a graph by using the method of Loukas (2019).

method to obtain the graph hierarchy, an example of which is shown in Figure 1. Then, our contribution is the learning of node representations for all graphs in this sequence. The high-level idea is to employ an encoder-decoder architecture: the encoder takes the graph and its node features as input and produces node features for the next graph in the hierarchy, whereas the decoder uses these produced node features to construct the graph. The objective is to obtain a decoding result as close to the given next graph as possible. The tool we use is variational Bayes. It is, however, slightly different from variational autoencoders (Kingma & Welling, 2014), because our decoder does not intend to reconstruct the input graph.

A clear benefit of unsupervised learning is that the learned representation is not tailored to a specific downstream task and hence may be more generalizable. Moreover, the coarsening method we adopt is a recent development that holds spectral guarantee (Loukas & Vandergheynst, 2018; Loukas, 2019) on the quality of the coarse graphs. We demonstrate the effectiveness of such a combination of hierarchical production and node representation learning in the context of graph classification. In particular, the classification performance is rather competitive with state-of-the-art supervised GNN approaches.

## 2 RELATED WORK

This work is in part based on graph coarsening that produces a hierarchy for a given graph. Denote by $G = (V, E)$ a graph with the vertex set $V$ and the edge set $E$. Graph coarsening is concerned with computing a smaller (coarse) graph $G_c = (V_c, E_c)$ with $|V_c| < |V|$ that retains the structure of $G$. A multilevel coarsening technique recursively coarsens the graph, yielding a hierarchy. Graph coarsening has been studied in the context of graph partitioning (Karypis & Kumar, 1998; Dhillon et al., 2007), graph visualization (Harel & Koren, 2000), machine learning (Lafon & Lee, 2006; Gavish et al., 2010; Ubaru & Saad, 2019), and pooling in graph neural networks (Bruna et al., 2014; Defferrard et al., 2016; Simonovsky & Komodakis, 2017). A variety of heuristic coarsening techniques have been proposed in different disciplines, including matching (Hendrickson & Leland, 1995; Ubaru & Saad, 2019), first choice (Cong & Shinnerl, 2013), contraction-based schemes (Dhillon et al., 2005; Sanders & Schulz, 2011), and algebraic multigrid (AMG)-inspired schemes (Sharon et al., 2000; Hu & Scott, 2001; Ron et al., 2011; Chen & Safro, 2011). Many well-known software packages exist for graph coarsening, e.g., Jostle (Walshaw & Cross, 2007), Metis (Karypis & Kumar, 1998), and DiBaP (Meyerhenke et al., 2008).

Recently, a few graph coarsening techniques achieving certain theoretical guarantees were presented (Moitra, 2011; Dorfler & Bullo, 2012; Loukas & Vandergheynst, 2018). Loukas (2019) presented variational approaches for graph coarsening with spectral guarantees. In particular, it was shown that the coarse graphs preserve the top eigenspace (whose dimension is an input to the method) within a predefined error tolerance. Here, we use this variational approach to obtain the graph hierarchy.

This work is concerned with unsupervised graph representation learning. Recent literature has focused on generative models to achieve the same; see. e.g., Kipf & Welling (2016); Li et al. (2018); Ma et al. (2018); Simonovsky & Komodakis (2018); Zhang et al. (2019). For learning hierarchical representations of graphs, most of the works that we are aware of are based on supervised learning, including Bruna et al. (2014); Defferrard et al. (2016); Simonovsky & Komodakis (2017); Ying et al. (2018b); Gao & Ji (2019); Lee et al. (2019). Methods most relevant to our work include: DIFFPOOL (Ying et al., 2018b), where the coarsening matrices are learned in an end-to-end fashion; GRAPH U-NET (Gao & Ji, 2019), where graph pooling is achieved using a learnable vector and

node ranking; and SAGPOOL (Lee et al., 2019), which is similar to GRAPH U-NET but uses graph self-attention to compute the ranking.

## 3 METHOD

The proposed method BAYESPOOL is an extension of variational autoencoders. As the name suggests, the goal of an autoencoder is to reconstruct the original input object after encoding it in the latent space. Our approach does not reconstruct the original input, but rather, aims as decoding an output faithful to another prescribed object. To this end, we first revisit variational Bayes and justify the use of variational lower bound for learning. Then, the machinery is applied to the graph context.

### 3.1 VARIATIONAL BAYES

Let $x$ be the observed (data) variable and $z$ be the unobserved (latent) variable. A core subject of Bayesian inference is concerned with estimating the posterior distribution $p(z|x)$. It is related to the prior $p(z)$ and the likelihood $p(x|z)$ through the Bayes theorem

$$p(z|x) = \frac{p(x|z)p(z)}{\int p(x,z)\,dz}.$$

The challenge lies in the marginalization over $z$ in the denominator, which is generally computationally intractable. Hence, various approximations were developed. Typically one adopts a surrogate model $q(z)$ independent of data; and recently in the context of VAEs, the data dependent distribution $q(z|x)$ is often used. In our setting, we introduce a new variable $\widetilde{x}$ and consider $q(z|\widetilde{x})$.

The difference, in terms of the Kullback–Leibler divergence, between the surrogate (variational) posterior $q(z|\widetilde{x})$ and the true posterior $p(z|x)$ may be decomposed as

$$
\begin{aligned}
D_{\mathrm{KL}}\Big(q(z|\widetilde{x})\,\|\,p(z|x)\Big) &= \int q(z|\widetilde{x})\log\frac{q(z|\widetilde{x})}{p(z|x)}\,dz \\
&= \underbrace{\int q(z|\widetilde{x})\log\frac{q(z|\widetilde{x})}{p(z)}\,dz}_{D_{\mathrm{KL}}\big(q(z|\widetilde{x})\,\|\,p(z)\big)} + \underbrace{\int q(z|\widetilde{x})\log p(x)\,dz}_{\log p(x)} - \underbrace{\int q(z|\widetilde{x})\log p(x|z)\,dz}_{\mathrm{E}_{q(z|\widetilde{x})}\big[\log p(x|z)\big]}.
\end{aligned}
$$

It consists of three terms: the KL divergence between the variational posterior and the prior $p(z)$, the log-evidence $\log p(x)$, and the marginal log-likelihood $\log p(x|z)$ under the surrogate distribution. Because the KL divergence is nonnegative, the log-evidence is lower bounded by the combination of the other two terms:

$$\log p(x) \geq \mathrm{E}_{q(z|\widetilde{x})}\Big[\log p(x|z)\Big] - D_{\mathrm{KL}}\Big(q(z|\widetilde{x})\,\|\,p(z)\Big). \tag{1}$$

The better the surrogate, the tigher the lower bound.

One sees that the right-hand side of (1) is almost the same as the usual log-evidence lower bound (ELBO), except that the surrogate $q(z|\widetilde{x})$ appears in place of $q(z|x)$. This observation is not surprising, because the marginalization is over the latent variable $z$ and has nothing to do with $x$ and $\widetilde{x}$. We thus conclude that the usual machinery of VAE applies, with only a notational change of the variational posterior. In the usual VAE setting, the first term of the right-hand side of (1) is considered the decoding accuracy, whereas the second term is a regularization in the latent space. Our setting follows this interpretation.

### 3.2 GRAPH REPRESENTATION LEARNING WITH VARIATIONAL BAYES

In our setting, a pair of graphs—the original one and the coarse one—is given. Let $A \in \mathbb{R}^{n \times n}$ and $A_c \in \mathbb{R}^{m \times m}$ be the corresponding graph adjacency matrices, respectively. Similarly, denote by $X \in \mathbb{R}^{n \times d}$ and $X_c \in \mathbb{R}^{m \times d'}$ the corresponding node feature matrices. We apply the encoder-decoder formalism, whereby the encoder encodes $A$ and $X$ into the coarse graph features $X_c$ that we seek, such that the decoder can use $X_c$ to decode a coarse graph as similar to $A_c$ as possible. See Figure 2 for an illustration.

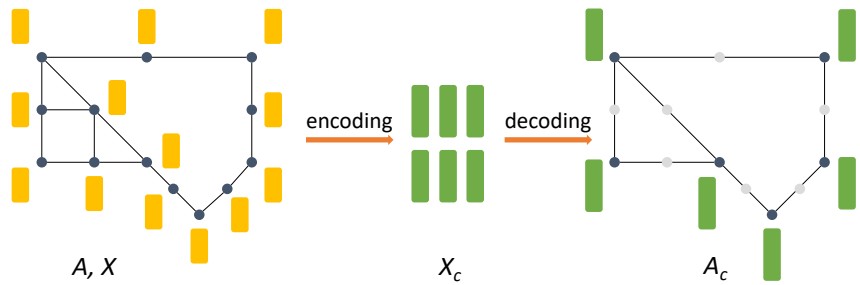

Figure 2: Encoder-decoder architecture.

Specifically, in the language of generative modeling, the encoder is the parameterized inference model that produces the parameters of $q(X_c|A, X)$ and the decoder is the parameterized generative model that produces the parameters of $p(A_c|X_c)$. Following (1), $A_c$ plays the role of $x$, $X_c$ plays the role of $z$, and $(A, X)$ plays the role of $\widetilde{x}$. The variational lower bound for model learning is thus

$$\text{ELBO} = \text{E}_{q(X_c|A,X)} \Big[ \log p(A_c|X_c) \Big] - D_{\text{KL}}\Big( q(X_c|A, X) \,\|\, p(X_c) \Big). \tag{2}$$

Maxmizing the ELBO amounts to maximizing the likelihood of decoding the coarse $A_c$ (given coarse node features $X_c$ resulting from the encoder), while minimizing a regularization of the variational posterior $q(X_c|A, X)$ that departs from the latent distribution $p(X_c)$.

### 3.3 MODELING AND PARAMETERIZATION

Generally, the latent space may be kept simple and unparameterized, with more emphasis placed on the encoder and the decoder. Thus, we let the prior $p(X_c)$ be the standard matrix normal $\mathcal{MN}(X_c \mid 0_{m \times d'}, I_m, I_{d'})$. Occasionally, specifying simple Gaussian structures on $p(X_c)$ may improve performance, such as letting $p(X_c)$ be the factored Gaussian, with the mean and the diagonal variance being parameters to learn. We have not yet, however, obtained strong empirical evidence of the benefit of using a parameterized factored Gaussian in this case.

For the decoder (generative model), a natural choice is to treat each element of the coarse graph adjacency matrix as a Bernoulli variable (scaled to the magnitude of the corresponding edge weight), with the success probability parameterized by a function of the corresponding coarse node features. For notational convenience, let the column vector $x_i^c \equiv X_c(i, :)^T$ and let $A_{ij}^c \equiv A_c(i, j)$. Then, $p(A_c|X_c)$ is the product of independent Bernoulli distributions:

$$p(A_c|X_c) = \prod_{i \neq j} p(A_{ij}^c \mid x_i^c, x_j^c) = \prod_{i \neq j} \text{Bernoulli}(\mathbf{1}_{A_{ij}^c} \mid p_{ij}),$$

where $\mathbf{1}_{A_{ij}^c}$ is the indicator function that returns 1 if $A_{ij}^c \neq 0$ and 0 otherwise, and $p_{ij}$ is a parameterized function that computes the success probability. A simple choice of the probability is an (unparameterized) dot product:

$$p_{ij} = \text{sigmoid}(\langle x_i^c, \ x_j^c \rangle), \tag{3}$$

but there exist several other straightforward parameterized variants. For example,

$$p_{ij} = \text{sigmoid}(\langle W^T x_i^c, \ W^T x_j^c \rangle) \quad \text{and} \quad p_{ij} = \text{sigmoid}(w^T(x_i^c \odot x_j^c)), \tag{4}$$

where $W$ and $w$ are a parameter matrix and vector, respectively. One may also replace them by an MLP. Note that all these functions are symmetric with respect to $i$ and $j$ because the graph is undirected.

For the encoder (inference model), we treat the variational posterior $q(X_c|A, X)$ as a factored Gaussian: each coarse node $x_i^c$ is an independent Gaussian with vector mean $\mu_i$ and diagonal variance $\text{diag}(\sigma_i^2)$. Then,

$$q(X_c|A, X) = \prod_i q(x_i^c|A, X) = \prod_i \mathcal{N}(x_i^c \mid \mu_i, \text{diag}(\sigma_i^2)),$$

where $\mu_i$ and $\sigma_i$ are parameterized functions of $A$ and $X$. By doing so, the KL term in the ELBO (2) admits a closed form. We transpose the column vectors $\mu_i$ and stack them to form a matrix $M$. Similarly, we proceed with the $\sigma_i$'s and form a matrix $S$. In what follows, we model the parameterized function for $M$. The one for $S$ is analogous.

Let $C$ be the set of coarse nodes; hence $A(C, :)$ keeps only the rows of $A$ that correspond to the coarse nodes. We let

$$M = \sigma(HXW_1), \tag{5}$$

where $W_1$ is a parameter matrix, $H$ has the same nonzero pattern as $A(C, :)$, and $\sigma$ is an activation function. This expression is in form similar to one graph convolution layer in GCN (Kipf & Welling, 2017), except that the square normalized adjacency matrix is replaced by a fat rectangular matrix $H$. Rather than basing $H$ on the original adjacency matrix, we designate its nonzero elements to be self-attention weights computed from the node features. Specifically, the nonzero elements of the $i$-th row of $H$ is computed as

$$\underset{j \in \text{neighbor}(i)}{\text{softmax}} \left( w_2^T \tanh(W_3 x_i + W_4 x_j) \right), \tag{6}$$

where $w_2$ is a parameter vector and $W_3$ and $W_4$ are parameter matrices. As an alternative, one may replace the attention calculation by that in GAT (Veličković et al., 2018):

$$\underset{j \in \text{neighbor}(i)}{\text{softmax}} \left( \text{LeakyReLU}(w_2^T [W_3 x_i; W_3 x_j]) \right). \tag{7}$$

To further enhance the representational power, in the parameterization (5) one may replace the original feature matrix $X$ by the node embedding matrix $Z$ output from GCN:

$$M = \sigma(HZW_1) \quad \text{with} \quad Z = \text{GCN}(A, X; W_5). \tag{8}$$

The GCN introduces additional parameters $W_5$ that may be useful for large data sets.

In Section 4, we experiment with the different variants and suggest a default choice that works generally well.

### 3.4 MULTILEVEL LEARNING

Coarsening may be done recursively, forming a sequence of increasingly coarse graphs. Let the adjacency matrices of this sequence be $A_0, A_1, \ldots, A_L$, where $A_0 = A$ corresponds to the initial given graph. Given this sequence and the initial feature matrix $X_0 = X$, the goal is to obtain the subsequent feature matrices $X_1, \ldots, X_L$.

To this end, model learning is conducted through maximizing the evidence of observing $A_1, \ldots, A_L$, treated independently. That is, we want to optimize $\log p(A_1) + \cdots + \log p(A_L)$. Following the argument as before, the actual quantity to optimize is the evidence lower bound. Inserting the layer index $\ell$ into (2), we have

$$\text{ELBO}_\ell = \text{E}_{q(X_\ell | A_{\ell-1}, X_{\ell-1})} \left[ \log p(A_\ell | X_\ell) \right] - D_{\text{KL}} \Big( q(X_\ell | A_{\ell-1}, X_{\ell-1}) \, || \, p(X_\ell) \Big).$$

Then, the log-evidence lower bound is the sum

$$\text{ELBO} = \sum_{\ell=1}^{L} \text{ELBO}_\ell.$$

The encoder and decoder parameters across coarsening levels differ but they are jointly learned through maximizing the ELBO.

### 3.5 DOWNSTREAM TASKS

The learned features $X_1, \ldots, X_L$, together with the original $X_0$, may be used for predictive tasks through learning a separate predictive model. We follow a common practice and define the model as

$$y_p = \text{MLP}\big(\text{concat}\big(\text{readout}(X_0), \text{readout}(X_1), \cdots, \text{readout}(X_L)\big)\big),$$

where readout is a global pooling across graph nodes (e.g., a concatenation of the max pooling and the mean pooling), concat denotes vector concatenation, MLP is self-explanatory, and $y_p$ is the class probability vector. In this paper, we consider the graph classification task.

## 4 EXPERIMENTS

In this section, we evaluate the performance of BAYESPOOL through the task of graph classification. Note again that BAYESPOOL is an unsupervised method; but as we will see, it is rather competitive with recently proposed supervised methods, even outperforming them on several data sets. We first present the details of the experimented data sets, the compared methods, and the training procedure. Then, we compare the graph classification accuracies. We also perform sensitivity analysis regarding the number of coarsening levels and compare the performance of several variants in the implementation of BAYESPOOL.

**Data sets:**  We consider the same data sets used by Lee et al. (2019). They are standard benchmarks publicly available from Kersting et al. (2016). Table 1 summarizes the information.

Table 1: Data set statistics.

| Data set | No. graphs | Nodes (max) | Nodes (avg) | Edges (avg) | Classes |
|---|---|---|---|---|---|
| DD | 1178 | 5748 | 284.32 | 715.66 | 2 |
| PROTEINS | 1113 | 620 | 39.06 | 72.82 | 2 |
| NCI1 | 4110 | 111 | 29.87 | 32.30 | 2 |
| NCI109 | 4127 | 111 | 29.68 | 32.13 | 2 |
| FRANKENSTEIN | 4337 | 146 | 16.90 | 17.88 | 2 |

The first two data sets are related to protein structures. The graphs in DD (Dobson & Doig, 2003) have different amino acids as nodes; and the edges correspond to the distance between the nodes. Labels indicate if the protein is an enzyme or not. The PROTEINS (Dobson & Doig, 2003) graphs have secondary structure elements of proteins as nodes. The edges indicate whether the nodes are in amino acids. NCI1 and NCI109 are biological data sets popularly used for anticancer activity classification (Wale et al., 2008). Here, the graphs correspond to chemical compounds, with the atoms and the chemical bonds represented as the nodes and edges, respectively. The FRANKEN-STEIN (Orsini et al., 2015) data set contains molecular graphs with 780 node features. The labels indicate if a molecule is mutagen or non-mutagen. Data sets DD, NCI1, and NCI109 do not come with node attributes for use as features. Hence, we employ transformations of node degrees as features, following the practice of DIFFPOOL.

**Compared methods:**  We compare with three supervised hierarchical graph representation learning methods, namely DIFFPOOL, GRAPH U-NET, and SAGPOOL. We duplicate the test accuracies of these methods reported by Lee et al. (2019).

DIFFPOOL (Ying et al., 2018b) computes hierarchical representations of graphs by using end-to-end trainable pooling based on soft clustering. This method is expensive because of the computation of the dense projection matrix. Lee et al. (2019) report that the method ran out of memory for a pooling ratio greater than 0.5. GRAPH U-NET (Gao & Ji, 2019) uses a learnable scoring vector to rank the graph nodes and selects top-ranked nodes for pooling. SAGPOOL (Lee et al., 2019) uses a similar approach as GRAPH U-NET for pooling, but incorporates a self-attention layer for learning the scoring vector. In both methods, the pooling ratio was set to be 0.5.

**Training procedure:**  We follow Lee et al. (2019) and perform several rounds of random splits with 80% training, 10% validation and 10% test. The learning rate is tuned over the range [1e-2, 5e-2, 1e-3, 5e-3, 1e-4]. The hidden dimensions are tuned over [10, 20, 32, 48]. For LeakyReLU, the slope coefficient is set to 0.01. The coarsening ratio $\rho$ is experimented with [0.25, 0.5, 0.75]; see Section 4.2 for sensitivity analysis. The dimension of the top eigenspace to be preserved by the coarsening procedure is set to $K = 5$. We implement the method in PyTorch and use Adam as the optimizer. The training employs early stopping with a patience of 50 epochs. The training setting for other methods are reported in Lee et al. (2019).

For a fair comparison, we use the same MLP classifier as in Lee et al. (2019). For readout, we use mean pooling and max pooling and concatenate the two outputs. We then use 3 feedforward layers along with softmax for classification. The classifier is trained for 150 epochs.

For architecture variation, the following combination consistently achieves the best results: the decoder uses an unparameterized dot product (3), the encoder uses parameterization (5) with the attention matrix $H$ computed by using the GAT form (7). The classification results reported in Section 4.1 below follow this choice. The results of other variants are reported in the subsections that follow.

The code is available at https://anonymous.4open.science/r/a50d6411-55f7-4e24-8f6c-6eecee118ea0/.

## 4.1 GRAPH CLASSIFICATION

We compare the performance of BAYESPOOL with several high-performing supervised methods recently proposed. These methods are all hierarchical methods. Table 2 lists the average test accuracies. The pooling/coarsening ratio is 0.5 in all cases.

Table 2: Graph classification accuracies.

| Method | DD | PROTEINS | NCI1 | NCI109 | FRANKENSTEIN |
|---|---|---|---|---|---|
| DIFFPOOL | 66.95 | 68.20 | 62.32 | 61.98 | 60.60 |
| GRAPH U-NET | 75.01 | 71.10 | 67.02 | 66.12 | 61.46 |
| SAGPOOL | 76.45 | 71.86 | **67.45** | **67.86** | 61.73 |
| BAYESPOOL | **78.12** | **76.85** | 61.86 | 61.04 | **62.65** |

BAYESPOOL outperforms other methods on three out of five data sets. Its performance is also on par with DIFFPOOL on the other two data sets, although infereior to GRAPH U-NET and SAGPOOL. The graphs in DD and PROTEINS are relatively large; with a coarsening ratio 50% there does not seem to cause information loss. Hence, BAYESPOOL works rather appealingly. On the other hand, although the graphs are smaller in FRANKENSTEIN, the data set contains a large number (780) of features, which possibly dwarf the graph structure information. Therefore, all methods yield similar results (with ours slightly outperforming the others). Encouragingly, BAYESPOOL is an unsupervised method; hence, these results show that the method is highly competitive for downstream tasks such as graph classification. It enables incorporating sophisticated coarsening techniques such as Loukas (2019) for graph representation learning.

Note that the adjacency matrices of the graphs are typically sparse. BAYESPOOL leverages sparse matrix computation, as opposed to DIFFPOOL where the projection matrix does not have an a priori sparsity structure. The coarsening procedure used by BAYESPOOL is implemented in sparse matrix format, along with the calculations of the neural network. This implementation results in a lower time cost and space complexity.

## 4.2 EFFECT OF COARSENING RATIO

One of the key factors that affects the performance of BAYESPOOL is the amount of graph reduction (the ratio of the coarse graph size to the initial size), or equivalently, the number of coarsening levels. This ratio is an input parameter to the coarsening method of Loukas (2019) that we use. In Table 3, we evaluate the performance of BAYESPOOL on two data sets with respect to the coarsening ratio. These data sets have relatively large graphs so that aggressive coarsening is possible.

Table 3: Graph classification accuracies for different coarsening ratios.

| Coarsening ratio | DD | PROTEINS |
|---|---|---|
| $\rho = 0.75$ | 77.87 | 77.27 |
| $\rho = 0.5$ | 78.12 | 76.85 |
| $\rho = 0.25$ | 68.75 | 66.96 |

In the table, we report the results for three different levels of coarsening with ratio $\rho = m/n$, where $m$ is the number nodes in the coarse graph and $n$ the original graph. We observe that the performance of BAYESPOOL is relatively stable when $\rho \geq 0.5$, but degrades as $\rho$ becomes smaller. As we lower $\rho$, more and more nodes and edges are removed, causing significant loss of information. However,

even for $\rho = 0.25$, BAYESPOOL still yields comparable results to DIFFPOOL according to Table 2. We conclude that $\rho = 0.5$ appears to be the right tradeoff.

## 4.3 VARIANTS OF ARCHITECTURE

As discussed in Section 3, a few parameterizations of the encoder and the decoder are possible. In this subsection, we comprehensively study the different variants, with the aim of obtaining a combination that generally works well. The results are reported in Table 4.

Table 4: Graph classification accuracies for different architecture variants.

| Encoder | | |
|---|---|---|
| Variants | DD | PROTEINS |
| LeakyReLU + plain [ (7) + (5) ] | 78.12 | 76.85 |
| LeakyReLU + GCN [ (7) + (8) ] | 75.68 | 74.51 |
| tanh + plain [ (6) + (5) ] | 75.48 | 73.78 |
| tanh + GCN [ (6) + (8) ] | 74.80 | 73.52 |

| Decoder | | |
|---|---|---|
| Variants | DD | PROTEINS |
| Plain dot product (3) | 78.12 | 76.85 |
| Parameterized dot product (4) left | 71.06 | 73.58 |
| Parameterized dot product (4) right | 71.69 | 70.58 |
| Replace $W$ in (4) left by 2-layer MLP | 72.83 | 75.65 |

In the top part of Table 4 we compare the performance of four variants of the encoder output $M$ (as well as $S$). The variants include (i) the attention calculation and (ii) whether or not to apply GCN before attention. The former contains two versions (6) and (7) and the latter also admits two versions (5) and (8), hence four combinations in total.

From the table, we observe that the GAT approach with LeakyReLU yields a better performance; and interestingly, using additionally GCN for parameterization lowers the performance. The introduction of the additional parameters inside GCN does not seem helpful.

In the bottom part of Table 4 we compare the performance of four variants of the decoder output $p_{ij}$. Along with the three variants discussed in Section 3.3 (unparamterized (3) and matrix/vector-paramterized (4)), we also consider replacing the matrix $W$ in (4) by a 2-layer MLP.

From the table, we observe that the plain dot product performs the best for both data sets. Again, the introduction of additional parameters does not seem helpful; rather, the accuracies deteriorate. This observation is consistent for both the encoder and the decoder. It is possible that the use of many parameters adversely affects the performance on data sets of a scale that we experimented with here.

## 5 CONCLUSION

We have presented an unsupervised approach BAYESPOOL for hierarchical graph representation learning. Compared with supervised approaches, a clear benefit is that the learned representations are generalizable to different downstream tasks. BAYESPOOL consists of an encoder-decoder architecture and adopts variational Bayes for training, but it is different from standard VAEs in that it does not attempt to reconstruct the input graph; rather, the decoder aims at producing the next graph in the hierarchy. Together with the use of the graph coarsening approach of Loukas (2019), we perform empirical evaluations that show that the learned representations yield competitive classification accuracies with state-of-the-art supervised GNN methods.

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
