# OpenReview forum: "Unsupervised Hierarchical Graph Representation Learning with Variational Bayes"
_ICLR.cc/2020/Conference — Reject_

### Official Review · AnonReviewer3 · 2019-10-21
**Official Blind Review #3**

**Rating:** 3

**Review:**

Summary: This work proposes an unsupervised hierarchical graph representation learning method, named BayesPool. The method learns a coarsening sequence of graphs together with the corresponding node representations. The coarsening sequence is learned using the method in Loukas (2019). The node representations are learned using an encoder-decoder structure, where the encoder encodes a graph to coarsened node representations, and the decoder decodes the node representations to a coarsened graph. The adopted objective function is analogous to VAE, except that the decoder does not aims to reconstruct an identical graph. Experiments on graph classification is performed on 5 different datasets, and competitive accuracy is achieved.

Concerns: The authors claim that the leant representation in an unsupervised manner is more desirable in terms of generalization. However, they only provide very limited experimental results, which is not very convincing. Moreover, the authors also do not explain clearly on when the node representation of the coarsening sequence is needed.

**Experience Assessment:**

I have published one or two papers in this area.

**Review Assessment: Checking Correctness Of Derivations And Theory:**

I carefully checked the derivations and theory.

**Review Assessment: Checking Correctness Of Experiments:**

I carefully checked the experiments.

**Review Assessment: Thoroughness In Paper Reading:**

I read the paper at least twice and used my best judgement in assessing the paper.

---

> ### Author Response · Authors · 2019-11-15
> **Response to Official Blind Review #3**
>
> Thank you very much for raising the concerns. We address them below. Hope the response helps you reassess the contribution of the work.
>
> RE: The point of unsupervised learning.
>
> The power of unsupervised representation comes from the fact that it is trained without knowing any downstream task. The learning uses much less information. We believe it is fair to consider that the method is successful, if the performance of a downstream task is comparable with that resulting from learning with additional supervised information.
>
> RE: When is the node representation of the coarsening sequence needed?
>
> The use of node features (both those of the original graph and the learned ones of the coarse graphs) is explained in section 3.5. Specifically, for each graph in the sequence, the node features are pooled to form a graph embedding, such that all graph embeddings can be concatenated to form the final graph representation. A simple predictive model (e.g., MLP) is trained separately for the downstream task (e.g., graph classification). This predictive model will vary depending on the nature of the task.

---

### Official Review · AnonReviewer1 · 2019-10-23
**Official Blind Review #1**

**Rating:** 6

**Review:**

The authors propose in this paper a new unsupervised graph representation learning method. The method leverages recent advances in graph coarsening, mainly Loukas' method. The key idea of the method consists in using a reconstruction target that is not the classical one in an auto-encoder setting. More precisely, the encoder takes as an input the original adjacency matrix and node features but the decode only aims at reconstructing the coarse adjacency matrix (obtained via Loukas' method).

The experimental evaluation is quite thorough and shows that the method performs quite well, especially considering it is unsupervised but is compared to supervised representation methods. It would be nice to include statistical tests to assess the significance of the differences in cases were accuracies are very close one to another. A missing part would be to explore the relevance of the learned representation for other tasks (i.e. to use a multi task data set). Of course as the representation is learned in an unsupervised way, one can argue that the current evaluation is already providing an answer.

Overall, I find the paper clear, but the variational bayes part could be much clearer. In fact I'm not sure why this is presented as variational bayes and not only variational. I do not see any prior distribution over parameters, for instance. I understand that the recent "tradition" in variational auto-encoder is to use this terminology, but as a (part time) bayesian, this is a bit annoying.

**Experience Assessment:**

I have read many papers in this area.

**Review Assessment: Checking Correctness Of Derivations And Theory:**

I did not assess the derivations or theory.

**Review Assessment: Checking Correctness Of Experiments:**

I carefully checked the experiments.

**Review Assessment: Thoroughness In Paper Reading:**

I made a quick assessment of this paper.

---

> ### Author Response · Authors · 2019-11-15
> **Response to Official Blind Review #1**
>
> Thank you very much for the comments. We respond to them in the following.
>
> RE: Significance of performance difference among methods.
>
> We well concur that significance tests are important for assessing the performance advantage of one model over another. On the other hand, we also note that the key message of this work is that unsupervised approaches may be as competitive as supervised approaches, even if no labels are used. We do not intend to claim that our approach is superior. In this regard, a very close accuracy exactly conveys this message.
>
> RE: Terminology regarding "variational Bayes".
>
> The terminology is subject to debate but we would like to share our opinion. It is not relevant to the contribution of this paper.
>
> Traditionally, the motivation for variational Bayes is approximate inference. The machinery was borrowed recently for developing generative models (VAE). More interesting in these models is the generative part (still, the mathematics is the same). The inference part serves only as a tool for training. The prior therein is an assumed distribution for the latent space; it is not used for parameters. In VAE, the parameters are with respect to the encoder network and the decoder network. Unless ones performs Bayesian deep learning, no priors on the network parameters exist for simplicity. On the other hand, the latent space is the prior for VAE. This prior may be made extremely simple (such as standard Gaussian), or slightly more expressive (such as a factored Gaussian parameterized by mean and variance), or even more complex. In this work, we find that the simple choice suffices.

---

> > ### Comment · AnonReviewer1 · 2019-11-15
> > **Variational Bayes**
> >
> > Again, I do not see the point in calling variational approximation variational Bayes is there is no prior on the parameters. You seem to be confusing variational approximation in e.g. EM where a complex distribution is replaced by a factored approximation with variational Bayes where the posterior distribution of the *parameters* is approximated by a factored distribution. Notice that using the Bayes rules does not turn a frequentist approach into a Bayesian one (eg the so-called naive Bayes classifier is not a Bayesian approach unless you add a prior on its parameter). And thus writing "A core subject of Bayesian inference is concerned with estimating the posterior distribution p(z|x)" where x is the observed data and z the latent variable is clearly misunderstanding of Bayesian inference.

---

### Official Review · AnonReviewer2 · 2019-10-29
**Official Blind Review #2**

**Rating:** 3

**Review:**

The paper proposes an unsupervised approach to learn a representation of graphs. The idea comes from an encoder-decoder architecture, which is common in related literature. The paper uses a variational Bayes approach in the learning process. Thorough experiments are provided to justify the feasibility of the method.

This paper provides an unsupervised style of learning graph representations, which may not be coupled with a specific downstream task so that it may be more useful in general; also, the experiments themselves seem to be at least comparable to the recent methods.

However, I vote for rejecting this submission for the following concerns.

(1) I did not find too many significant differences between this paper and [Kingma & Welling, 2014] in the design of encoder-decoder architecture as well as the learning procedure (I am not an expert in this area so please correct me if I am wrong).

(2) The intuition of learning the representation in an unsupervised manner is interesting and important to me, though the experiments are mostly on the classification tasks. I think it would be helpful to demonstrate the representation power of the learned representation of the graph in tackling other tasks.


**Experience Assessment:**

I do not know much about this area.

**Review Assessment: Checking Correctness Of Derivations And Theory:**

I assessed the sensibility of the derivations and theory.

**Review Assessment: Checking Correctness Of Experiments:**

I assessed the sensibility of the experiments.

**Review Assessment: Thoroughness In Paper Reading:**

I made a quick assessment of this paper.

---

> ### Author Response · Authors · 2019-11-15
> **Response to Official Blind Review #2**
>
> Thank you very much for raising the concerns. Our response is in the following. Hope the response help you reassess the contribution of the work.
>
> RE: Concern (1); difference from Kingma & Welling (2014).
>
> A few details distinguish our approach from the VAE work by Kingma & Welling:
>
> (a) Kingma & Welling laid down a variational principle for unsupervised learning but the neural architectures of encoder and decoder are not specified. A plethora of work emerges to address various data types, through designing specific encoders and decoders. In this work we focus on graph data.
>
> (b) In most (if not all) settings, an autoencoder tries to reconstruct the data itself. We, on the other hand, are concerned with constructing a different graph. This needs the adjustment of the theory and the loss function. We mathematically justify so in Section 3.1.
>
> (c) One may recall the variational graph autoencoder work by Kipf & Welling (2016). Despite under the same VAE umbrella, we need a different architecture to cope with the fact that a different graph is constructed, besides the adjustment of the variational Bayes formulation. Moreover, our work is concerned with graph-level representation rather than node-level. Hence, an additional component that forms the graph embedding from node features is needed.
>
> RE: Concern (2); unsupervised representation.
>
> The power of unsupervised representation comes from the fact that it is trained without knowing any downstream task. The learning uses much less information. We believe it is fair to consider that the method is successful, if the performance of a downstream task is comparable with that resulting from learning with additional supervised information.

---

### Decision · Program_Chairs · 2019-12-19

**Decision:**

Reject

**Comment:**

The paper presents an unsupervised method for graph representation, building upon Loukas' method for generating a sequence of gradually coarsened graphs. The contribution is an "encoder-decoder" architecture trained by variational inference, where the encoder produces the embedding of the nodes in the next graph of the sequence, and the decoder produces the structure of the next graph.

One important merit of the approach is  that this unsupervised representation can be used effectively for supervised learning, with results quite competitive to the state of the art.

However the reviewers were unconvinced by the novelty and positioning of the approach. The point of whether the approach should be viewed as variational Bayesian, or simply variational approximation was much debated between the reviewers and the authors.

The area chair encourages the authors to pursue this very promising research, and to clarify the paper; perhaps the use of "encoder-decoder" generated too much misunderstanding.
Another graph NN paper you might be interested in is "Edge Contraction Pooling for Graph NNs", by Frederik Diehl.